# Taste Responses and Ingestive Behaviors to Ingredients of Fermented Milk in Mice

**DOI:** 10.3390/foods12061150

**Published:** 2023-03-08

**Authors:** Yuko Yamase, Hai Huang, Yoshihiro Mitoh, Masahiko Egusa, Takuya Miyawaki, Ryusuke Yoshida

**Affiliations:** 1Department of Dental Anesthesiology and Special Care Dentistry, Graduate School of Medicine, Dentistry and Pharmaceutical Sciences, Okayama University, Okayama 700-8525, Japan; 2Department of Oral Physiology, Graduate School of Medicine, Dentistry and Pharmaceutical Sciences, Okayama University, Okayama 700-8525, Japan; 3Faculty of Medicine, Dentistry and Pharmaceutical Sciences, Okayama University, Okayama 700-8525, Japan; 4Advanced Research Center for Oral and Craniofacial Sciences, Okayama University Dental School, Okayama 700-8525, Japan

**Keywords:** postingestive effects, galactose, lactose, oligosaccharides, lactic acid

## Abstract

Fermented milk is consumed worldwide because of its nutritious and healthful qualities. Although it is somewhat sour, causing some to dislike it, few studies have examined taste aspects of its ingredients. Wild-type mice and T1R3-GFP-KO mice lacking sweet/umami receptors were tested with various taste components (sucrose, galactose, lactose, galacto-oligosaccharides, fructo-oligosaccharides, l- and d-lactic acid) using 48 h two-bottle tests and short-term lick tests. d-lactic acid levels were measured after the ingestion of d- or; l-lactic acid or water to evaluate d-lactic acidosis. In wild-type mice, for the sweet ingredients the number of licks increased in a concentration-dependent manner, but avoidance was observed at higher concentrations in 48 h two-bottle tests; the sour ingredients d- and l-lactic acid showed concentration-dependent decreases in preference in both short- and long-term tests. In 48 h two-bottle tests comparing d- and l-lactic acid, wild-type but not T1R3-GFP-KO mice showed higher drinking rates for l-lactic acid. d-lactic acidosis did not occur and thus did not contribute to this preference. These results suggest that intake in short-term lick tests varied by preference for each ingredient, whereas intake variation in long-term lick tests reflects postingestive effects. l-lactic acid may have some palatable taste in addition to sour taste.

## 1. Introduction

The foods that humans consume are diverse. Among them, fermented milk originated in the Middle East at least as early as 1000 B.C. and has long been consumed by people around the world [1] for both its nutritive and functional qualities, such as good preservability, good taste quality, and good effects on our health. Several studies have associated yogurt consumption with lower anthropometric indicators [2] and reduced risk of type 2 diabetes mellitus [3] and cardiovascular diseases [4].

One of the flavor characteristics of fermented milk is its strong sour taste, thought to be elicited by lactic acid. Of the two enantiomers, d- and l-lactic acid, l-lactic acid is a common compound of human metabolism, whereas d-lactic acid is produced by some strains of microorganisms or by some less relevant metabolic pathways [5]. The amount of total lactic acid in fermented milk products ranges from 0.6% to 1.2%, with l-lactic acid the major enantiomer [6]. For example, about 60% of the total lactic acid is the l-form enantiomer in yogurt [6]. Lactic acid is also involved in other fermented foods and drinks such as wine [7] and sake [8] and may contribute greatly to their taste. Sour taste is an innate aversive taste quality, and the strong sourness of fermented milk may discourage its intake. On the other hand, sourness has some beneficial functions for the body, such as increasing salivary secretion [9], and could improve health if used properly. Very few studies have investigated the beneficial effects of sourness. In *Drosophila*, lactic acid is an appetitive and energetic tastant since it stimulates feeding through the activation of sweet gustatory receptor neurons [10]. Taste substances are usually categorized to a single taste modality sensed by a single receptor or family of receptors. However, taste detection for some chemical species could be complex, with certain molecular properties acting differently on multiple receptors. For example, artificial sweeteners such as saccharin and cyclamate have been reported to activate bitter taste receptors along with sweet taste receptors [11]. It may be possible that in some animals lactic acid is detected by not only the sour taste receptor, otopetrin-1 (OTOP1) [12,13], but also by other receptors such as sweet receptors.

Fermented milk also contains sugars such as lactose and galactose and oligosaccharides. Recently, oligosaccharides such as galacto-oligosaccharides (GOSs) and fructo-oligosaccharides (FOSs) have been added to commercially available yogurt as prebiotics. Lactose is the major carbohydrate in breast milk, followed by oligosaccharides and glucose [14]. Lactose, galactose, and oligosaccharides are important nutritional sources during the suckling period in mammals, including humans. Infants fed breast milk or artificial milk containing lactose had higher levels of glucose and essential amino acids (leucine, isoleucine, valine, and proline) in their blood than those fed artificial milk without lactose [15]. However, the downregulation of lactase after infancy has also been found to cause lactose intolerance [16]. Fructo-oligosaccharides and GOSs act as prebiotics when added to artificial milk, bringing the intestinal microflora of artificially fed infants closer to the bifido-dominant microflora of breast-fed infants [17].

Taste receptors are expressed throughout the body; for example, sweet taste receptors in the gastrointestinal tract are involved in the enhancement of the absorption of glucose [18] and bitter taste receptors function in biological defense in the trachea [19]. Fermented milk ingredients may have meaningful functions for the body via taste sensors. If the functionality of these components derived from fermented milk can be elucidated, it will provide new motivation for the consumption of fermented milk, which may increase its consumption and contribute greatly to the maintenance and promotion of people’s health.

To elucidate the taste aspects of fermented milk, here we investigated taste responses and ingestive behaviors to the ingredients of fermented milk in mice. As a characteristic point, we used two types of licking tests. One was a short-term (5 s) lick test to purely examine taste responses to each ingredient. The other was a long-term (48 h) two-bottle test, which involved the post ingestive effect of each ingredient in addition to taste response. We chose galactose, lactose, GOSs, and FOSs as sweet ingredients derived from fermented milk, and d- and l-lactic acid as sour ingredients. In addition, preference for the two enantiomers d- and l-lactic acid was compared in wild-type (WT) mice and T1R3-GFP-KO mice lacking the sweet receptor component T1R3 and expressing green fluorescent protein (GFP). In 48 h two-bottle tests, WT mice showed the concentration-dependent increased intake of the sweet ingredients (but avoided higher concentrations), and both WT and T1R3-GFP-KO mice showed concentration-dependent decreases for the sour ingredients d- and l-lactic acid. The WT but not the T1R3-GFP-KO mice had higher intake rates for l- than for d-lactic acid; d-lactic acidosis did not occur and thus cannot explain the preference for l-lactic acid, suggesting l-lactic acid may have some palatable taste in addition to sour taste.

## 2. Materials and Methods

### 2.1. Ethical Approval

All animal experiments were performed in accordance with the National Institute of Health Guide for the Care and Use of Laboratory Animals and approved by the Committee for Laboratory Animal Care and Use and the local ethics committee at Okayama University, Japan.

### 2.2. Animals

These experiments used male C57BL/6J (WT) mice (purchased from CLEA Japan, Tokyo, Japan) and mice lacking the *Tas1r3* gene but expressing green fluorescent protein in cells that usually express T1R3 (T1R3-GFP-KO mice), generated by crossing T1R3-KO mice [20] and T1R3-GFP mice [21] originally generated at Mount Sinai Medical School from the C57BL/6J strain and maintained in this background. We used 54 WT mice and 18 T1R3-GFP-KO mice for 48 h 2-bottle tests, 7 WT mice and 8 T1R3-GFP-KO mice for short-term lick tests, and 9 WT mice for measurement of blood d-lactic acid levels. All mice were maintained in a 12/12 h light/dark cycle and fed standard rodent chow (MF, Oriental yeast co., Tokyo, Japan). Animals were 8–20 weeks of age, weighing 20–35 g.

### 2.3. Short-Term Lick Test

The WT mice (*n* = 7) and T1R3-GFP-KO mice (*n* = 8), housed in individual cages, were used as experimental subjects. On day 1 of training, each animal was water deprived for 12 h and then placed in the test cage and given free access to deionized water during the 1 h session. Days 2–5 were training sessions: animals were trained to drink deionized water on an interval schedule, consisting of 5 s periods of deionized water presentation alternating with 10 s intertrial intervals. From day 6, the numbers of licks for each taste solution and deionized water were counted during the first 5 s after the animal’s first lick, using a lick meter (Yutaka Electronics Co., Gifu, Japan). The test solutions used were 1–1000 mM sucrose + 1 mM quinine hydrochloride (QHCl), 1–1000 mM galactose + 1 mM QHCl, 1–444 mM lactose + 1 mM QHCl (444 mM is saturated lactose solution at 20 °C), 1–30% GOSs + 1 mM QHCl, 1–30% FOSs + 1 mM QHCl, 1–100 mM citric acid, 1–100 mM d-lactic acid, and 1–100 mM l-lactic acid. One tastant, at varying concentrations, was tested on any given test day. To examine lick responses to preferred solutions (sucrose, galactose, lactose, GOSs, and FOSs), mice were deprived of both food and water 12 h before the experiment and 1 mM QHCl was added to test solutions (bitter-sweet mixture) to obtain clear concentration-dependent preference to sweeteners [22]. On each test day, mice were given test solutions with concentrations of descending order (from highest concentration to deionized water) in first trial then randomized order in second and further trials. To examine lick responses to aversive solutions (citric acid and d- and l-lactic acid), mice were deprived of water 12 h before beginning of experiment. On each test day, mice were given test solutions in ascending concentration order (from deionized water to highest concentration) in first trial and then randomized order in second and further trials. The number of lick trials for each solution was at least three, and their values were averaged for data analysis. l- and d-lactic acid were purchased from Musashino Chemical Laboratory (Tokyo, Japan); other ingredients were purchased from Nakarai Tesque (Kyoto, Japan) or FUJIFILM Wako Pure Chemical Corporation (Osaka, Japan).

### 2.4. Long-Term Intake Test: The 48 Hour 2-Bottle Test

The WT mice (9 groups, 6 mice each) and T1R3-GFP-KO mice (3 groups, 6 mice each) were used as experimental subjects. Mice were caged individually and given access for 48 h to two bottles, one containing deionized water and the other containing the test solution. The bottles were inserted into the individual’s home cage, and their positions were switched after 24 h to avoid location preference.

Total intake of each solution after 48 h was measured, and the preference ratio (PR) was calculated as the amount of ingested test solution divided by the total amount ingested (water + test solution). Preference ratio > 0.5 indicated preference; PR < 0.5 indicated avoidance.

For each tastant, presentation was within an ascending concentration series. The test solutions used were 10–1000 mM sucrose, 10–1000 mM galactose, 10–444 mM lactose, 1–30% GOSs, 1–30% FOSs, 1–100 mM citric acid, 1–100 mM l-lactic acid, 1–100 mM d-lactic acid.

For comparison between d- and l-lactic acid, mice were given access for 48 h to two bottles, one each containing d- or l-lactic acid at the same concentration. Preference ratio for l-lactic acid was calculated as the amount of ingested l-lactic acid divided by the total amount of ingested solution (l- + d-lactic acid).

### 2.5. Measurement of Blood d-Lactic Acid

Male WT mice (*n* = 9), housed in individual cages, were used as experimental subjects. On test day 1, at around 8 pm mice were weighed and placed in individual cages with food and one bottle containing 30 mM l-lactic acid, 30 mM d-lactic acid, or deionized water. After 12 h of presentation, body weight and intake of food and solution were measured for each mouse and a blood sample was collected from the tail vein. On days 3 and 5, this procedure was repeated with a different solution; on days 2 and 4, mice were maintained in their home cage with food and water. Mice were divided into three groups (*n* = 3 each); for each group, solutions were presented with the same order (deionized water, l-lactic acid, or d-lactic acid) but starting with a different solution, to test for order effects.

Blood d-lactic acid concentration was measured using ELISA (PicoProve^TM^ d-Lactate Fluorometric Assay Kit, BioVision, Boston, MA, USA) following the manufacturer’s instructions.

### 2.6. Statistical Analysis

Prior to testing, the Shapiro–Wilk test was used to check whether each variable followed a normal distribution.

For short-term lick tests, and long-term 48 h 2-bottle tests, differences among concentrations of each ingredient were statistically analyzed by one-way ANOVA and post hoc Tukey highest-significant-difference (HSD) test. Differences among taste ingredients (GOSs vs. FOSs, d-lactic acid vs. l-lactic acid (vs. CA)) and concentration were statistically analyzed by two-way ANOVA. Differences among species (WT vs. T1R3-GFP-KO) and concentration were statistically analyzed by two-way ANOVA. Differences in blood d-lactic acid levels, weight changes, and amount of feeding and drinking were statistically analyzed by one-way ANOVA.

Statistical analyses were performed using EZR software [23]. A *p*-value < 0.05 was considered significant.

## 3. Results

### 3.1. Taste Response and Ingestive Behaviors to Sugars in WT Mice

In the short-term lick test, WT mice showed a concentration-dependent increase in the number of licks of sucrose+quinine (F = 137.6, *p* < 0.001), lactose+quinine (F = 30.3, *p* < 0.001), and galactose+quinine solution (F = 128, *p* < 0.001; one-way ANOVA) (Figure 1A), suggesting strong taste preference for these sugars at high concentrations. In the 48 h two-bottle test, WT mice also showed a concentration-dependent increase in PR for sucrose (F = 74.7, *p* < 0.001, one-way ANOVA); although PRs for lactose and galactose solutions at ≤300 mM increased in a concentration-dependent manner (lactose: F = 36.7, *p* < 0.001; galactose: F = 21.2, *p* < 0.001; one-way ANOVA), mice apparently avoided the highest concentration of lactose (10 mM vs. 444 mM: *p* < 0.001, post hoc Tukey HSD test) and showed decreased preference for the highest concentration of galactose (10 mM vs. 1000 mM: *p* = 0.419, post hoc Tukey HSD test) (Figure 1B). It is worth noting that calorie content and sugar concentration are not matched on galactose versus lactose and sucrose.

### 3.2. Taste Response and Ingestive Behaviors to Oligosaccharides in WT Mice

In the short-term lick test, WT mice showed a concentration-dependent increase in the number of licks of solutions of GOSs+quinine (F = 35.6, *p* < 0.001) and FOSs+quinine (F = 173.7, *p* < 0.001; one-way ANOVA) (Figure 2A). The WT mice showed a stronger taste preference for FOSs than for GOSs at the same concentration (effect of ingredient, F(_1,60)_ = 49.65, *p* <0.001, two-way ANOVA). In the 48 h two-bottle test, WT mice showed increased preference for GOSs (F = 24.3, *p* < 0.001) and FOSs (F = 24.0, *p* < 0.001; one-way ANOVA) at ≤10% but avoided them at the highest concentration (30%) (GOSs: 1% vs. 30%, *p* < 0.001; FOSs: 1% vs. 30%, *p* = 0.006; post hoc Tukey HSD test) (Figure 2B). Mice were likely to ingest more FOSs than GOSs at the same concentration (effect of ingredient, F_(1,40)_ = 16.63, *p* < 0.001, two-way ANOVA).

### 3.3. Taste Responses and Ingestive Behaviors to Acids in WT Mice

Both in the 48 h two-bottle test and in the short-term lick test, WT mice showed a concentration-dependent avoidance to citric acid (short-term lick test: F = 76.3, *p* < 0.001; two-bottle test: F = 52.6, *p* < 0.001), d-lactic acid (short-term lick test: F = 524.2, *p* < 0.001; two-bottle test: F = 2.3, *p* = 0.086), and l-lactic acid solution (short-term lick test: F = 192.3, *p* < 0.001; two-bottle test: F = 31.5, *p* < 0.001; one-way ANOVA) (Figure 3A,B). We found no statistically significant differences among these acids in numbers of licks (effect of ingredient, F_(2,90)_ = 0.93, *p* = 0.40) or PR (effect of ingredient, F_(2,75)_ = 0.41, *p* = 0.67, two-way ANOVA).

We also compared preference for enantiomers of lactic acid (Figure 3C). At the lowest concentration (1 mM), the PR for l-lactic acid was almost chance level (50%). The WT mice showed no preference or avoidance for l- over d-lactic acid at 3–10 mM but a preference for l- over d-lactic acid at 30 and 100 mM (F = 4.74, *p* < 0.01, one-way ANOVA). Post hoc tests showed a statistically significant difference between 1 and 30 mM (*p* < 0.05) and between 1 and 100 mM (*p* < 0.05) (Figure 3C).

### 3.4. Preference for l- over d-Lactic Acid in T1R3-GFP-KO Mice

Why did mice prefer l-lactic acid over d-lactic acid at the same concentration? We hypothesized that l-lactic acid, unlike d-lactic acid, might include a more “palatable” taste quality, such as sweet or umami, since different tastes for l- and d-amino acid have been reported [24]. To test this, we used mice lacking the sweet and umami receptor component T1R3 (T1R3-GFP-KO mice).

In short-term lick tests, T1R3-GFP-KO mice did not show any increase in the number of licks for sugars+quinine (Figure 4A) or oligosaccharides+quinine (Figure 4B) although they licked well water (34.5–36.7 licks/5 s), suggesting that T1R3-GFP-KO mice have no or reduced taste sensitivity to these sweet compounds, due to a lack of the sweet receptor component T1R3. Similar to WT mice, T1R3-GFP-KO mice showed concentration-dependent avoidance to d- and l-lactic acid in the short-term lick test (l-lactic acid: F = 145, *p* < 0.001; d-lactic acid: F = 182.7, *p* < 0.001; one-way ANOVA). The same tendency was observed in the 48 h two-bottle test (l-lactic acid: F = 5.89, *p* < 0.01; d-lactic acid: F = 17.42, *p* < 0.001; one-way ANOVA) (Figure 4C,D). There was no significant difference between d- and l-lactic acid in the number of licks (effect of ingredient, F(1, 56) = 2.69, *p* = 0.11) or PR (effect of ingredient, F(1,50) = 3.04, *p* = 0.09; two-way ANOVA) (Figure 4C,D). Unlike WT mice, T1R3-GFP-KO mice showed no preference for l- over d-lactic acid at any concentration (F = 0.87, *p* = 0.5, one-way ANOVA) (Figure 4E), suggesting that T1R3-GFP-KO mice could not distinguish between l- and d-lactic acid.

We also compared the number of licks in the short-term lick test between WT and T1R3-GFP-KO mice (Figure 5). The number of licks to sugars and oligosaccharides were significantly different between WT and T1R3-GFP-KO mice (Figure 5A–E, Table 1). However, the number of licks to l- and d-lactic acid were not significantly different between WT and T1R3-GFP-KO mice (Figure 5F,G, Table 1). Because of the lack of T1R3, sensitivity to sugars and oligosaccharide, which have a strong sweet taste at higher concentration, may be different between WT and T1R3-GFP-KO mice.

### 3.5. Blood d-Lactic Acid Levels after Ingestion of d- and l-Lactic Acid

The WT mice preferred l- over d-lactic acid in the 48 h two-bottle test (Figure 3C). This preference may be due to d-lactic acidosis, which might occur by drinking d-lactic acid; mice might thus show aversion to d-lactic acid, leading them to drink more l- than d-lactic acid in the 48 h two-bottle test. To test this possibility, we measured blood d-lactic acid levels after mice drank d- or l-lactic acid or water for 12 h (Figure 6). At 12 h after ingestion, there were no statistically significant differences in weight change (Figure 6A; *p* = 0.699), total amount of food intake (Figure 6B; *p* = 0.159), or total amount of drinking solution (Figure 6C; *p* = 0.082; one-way ANOVA), and blood d-lactic acid levels were almost similar among the three groups (Figure 6D; *p* = 0.949, one-way ANOVA), indicating that drinking d-lactic acid did not lead to d-lactic acidosis. Taken together, these data suggest that l-lactic acid may have some preferable taste detected by T1R3-dependent sweet/umami receptors in mice.

## 4. Discussion

In this study, we examined taste responses to components of fermented milk in mice. We demonstrated that WT mice but not T1R3-GFP-KO mice preferred galactose, lactose, GOSs, and FOSs—all sweet components in fermented milk—in a concentration-dependent manner in the short-term lick test (Figure 1A, Figure 2A, Figure 3A,B) but showed avoidance of these ingredients at higher concentrations in the long-term, 48 h two-bottle preference test (Figure 1B and Figure 2B). The WT mice showed similar avoidance to both l- and d-lactic acid—sour ingredients of fermented milk—in both short-term lick tests and long-term preference tests (Figure 3A,B) but preferred l- over d-lactic acid in the long-term test (Figure 3C). Although T1R3-GFP-KO mice also showed similar avoidance to both l- and d-lactic acid in both short-term lick tests and long-term preference tests (Figure 4C,D), they did not prefer l- over d-lactic acid in the long-term test (Figure 4E). These results suggest that l-lactic acid but not d-lactic acid may be detected by taste receptors containing T1R3, inducing a sweet or umami taste in addition to a sour taste. We also examined the postingestive effect of d-lactic acidosis but found no significant increase in blood d-lactic acid levels after the ingestion of d-lactic acid (Figure 6D), suggesting that d-lactic acidosis may not contribute to preference for l- over d-lactic acid. To the best of our knowledge, this is the first study to comprehensively examine the taste responses to fermented milk components in a behavioral experiment.

The WT mice showed avoidance to galactose, lactose, GOSs, and FOSs at high concentrations in the long-term preference test, possibly due to the negative postingestive effect of these sweet ingredients. Galactose, produced by the digestion of lactose and GOSs, has been found to bind to sodium glucose transporter 1 (SGLT1) in the small intestine and cause postprandial effects, and mice showed a concentration-dependent preference for galactose but decreased preference for a 16% galactose solution [25]. In the present study, we used galactose at 1000 mM (~18% solution) and 300 mM (~5.4%). Thus, our results are consistent with this previous study. Lactose is an important carbohydrate in weaning and is mainly digested by lactase in the small intestine into galactose and glucose before being used by the organism [26]. In an earlier study, weaning rats preferred low-lactose (12%) over high-lactose (47%) liquid diets because of negative postingestive effects [27]. In the present study, we used adult mice (8–20 weeks of age) and used lactose at 444 mM (~16% solution) and 300 mM (~10.8%). Thus, our results are similar to this previous study.

Mice learn to associate a novel taste with poor physical condition and, as a consequence, avoid drinking fluid with this specific taste; this is conditioned taste aversion [28]. Lactose is digested by lactase. Most mammals, including humans, have very high amounts of latent lactase during suckling, but after weaning the amount of lactase decreases [29]. This leads to lactose malabsorption, which may induce gastrointestinal symptoms such as bloating, borborygmi, flatulence, abdominal pain, and diarrhea [30]. Therefore, mice might associate the ingestion of high concentrations of lactose with gastrointestinal symptoms and then avoid drinking lactose solutions. Galacto-oligosaccharides consist of 2–8 monomeric units containing galactose and glucose; FOSs consists of 3–5 monomeric units containing fructose and glucose. Both GOSs and FOSs are not digested by digestive enzymes but are known to be degraded by intestinal bacteria such as *Bifidobacterium* [31]. Therefore, it is possible that the galactose produced by the breakdown of GOSs by intestinal bacteria might induce a negative postingestive effect, leading to the avoidance of GOSs in long-term tests. However, fructose does not have a negative postingestive effect, and glucose has only a positive postingestive effect [25]. Thus, the breakdown of FOSs by intestinal bacteria may contribute not to avoidance but to preference for FOSs. Therefore, similar to lactose, the maldigestion of GOSs and FOSs may lead to gastrointestinal symptoms [32], and then mice avoid intake of solutions containing GOSs and FOSs at higher concentrations.

Why did WT mice prefer l-lactic acid compared to d-lactic acid at high concentrations in long-term preference tests? One possible reason may be the sweet or umami taste of l-lactic acid: T1R3-GFP-KO mice showed no preference for l- over d-lactic acid at high concentrations (Figure 4E). In *Drosophila*, lactic acid is an appetitive and energetic tastant, and it stimulates feeding through the activation of sweet gustatory receptor neurons [10]. Human studies have shown that there are significant differences in taste among enantiomers of amino acids. For example, l-leucine has a bitter taste, while d-leucine has a sweet taste [24]. The enantiomers of lactic acid may have a similar effect on taste receptors. We hypothesize that T1R3, a component of “highly palatable” sweet and umami taste receptors, could detect l-lactic acid but not d-lactic acid, leading to a preferable taste of l- over d-lactic acid. This possibility should be investigated in future studies.

The second possibility is that drinking d-lactic acid may lead to d-lactic acidosis in mice. The standard metabolism of l-lactic acid in most organisms is mediated by l-lactate dehydrogenase, which is involved in a basic metabolism tightly linked to glycolysis and gluconeogenesis, and it is a crucial part of the Cori cycle in humans and other higher mammals [5]. In contrast, d-lactic acid is known as a harmful enantiomer and can cause d-lactic acidosis if excess d-lactate is not fully metabolized by d-lactate dehydrogenase [5]. However, we examined d-lactic acid levels in blood after mice ingested d- or l-lactic acid or water and found almost similar levels (Figure 6D). Thus, d-lactic acidosis may not contribute to the preference for l- over d-lactic acid.

The third possibility is that wild-type mice may be better able to metabolize l-lactic acid than T1R3-GFP-KO mice. Sweet taste receptors are known to be involved in digestion and absorption in the intestine [18]. Such sweet receptors in the gastrointestinal tract may be involved in the metabolism of l-lactic acid, inducing some positive postingestive effect in WT mice but not in T1R3-GFP-KO mice. Such a possibility should also be investigated in future studies.

In summary, we demonstrated that mice showed avoidance to sweet ingredients in fermented milk at high concentrations in 48 h two-bottle tests even though they prefer these ingredients in short-term lick tests. We also found that mice preferred l-lactic acid over d-lactic acid, both of which are sour ingredients in fermented milk. The avoidance of sugars and oligosaccharides at high concentrations may be due to the postingestive effects of the sweet ingredients. l-lactic acid may have some sweet or umami taste in addition to sour taste, leading to a preference for l-lactic acid over d-lactic acid.

## Figures and Tables

**Figure 1 foods-12-01150-f001:**
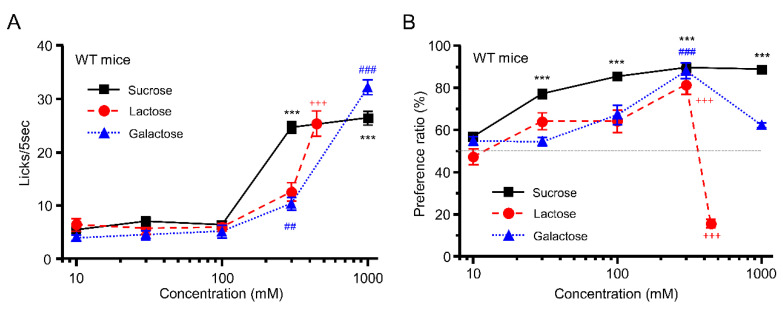
Behavioral responses to sugars in wild-type (WT) mice. (**A**) Number of licks of sucrose (black squares), lactose (red circles), and galactose (blue triangles) in the short-term (5 s) lick test (*n* = 7). (**B**) Preference ratio for sucrose (black squares), lactose (red circles), and galactose (blue triangles) in the 48 h two-bottle test (*n* = 6). ^##^
*p* < 0.01, *** *p* < 0.001, ^+++^
*p* < 0.001, ^###^
*p* < 0.001, post hoc Tukey highest-significant-difference (HSD) test (vs. 10 mM). All data are presented as the mean ± standard error.

**Figure 2 foods-12-01150-f002:**
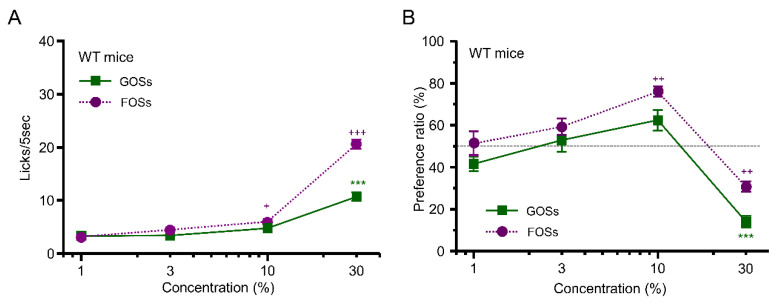
Behavioral responses to oligosaccharides in WT mice. (**A**) The number of licks of galacto-oligosaccharides (GOSs, green squares) and fructo-oligosaccharides (FOSs, purple circles) in the short-term (5 s) lick test (*n* = 7). (**B**) Preference ratio for GOSs (green squares) and FOSs (purple circles) in the 48 h two-bottle test (*n* = 6). + *p* < 0.05, ++ *p* < 0.01, *** *p* < 0.001, +++ *p* < 0.001, post hoc Tukey HSD test (vs. 1%). All data are presented as the mean ± standard error.

**Figure 3 foods-12-01150-f003:**
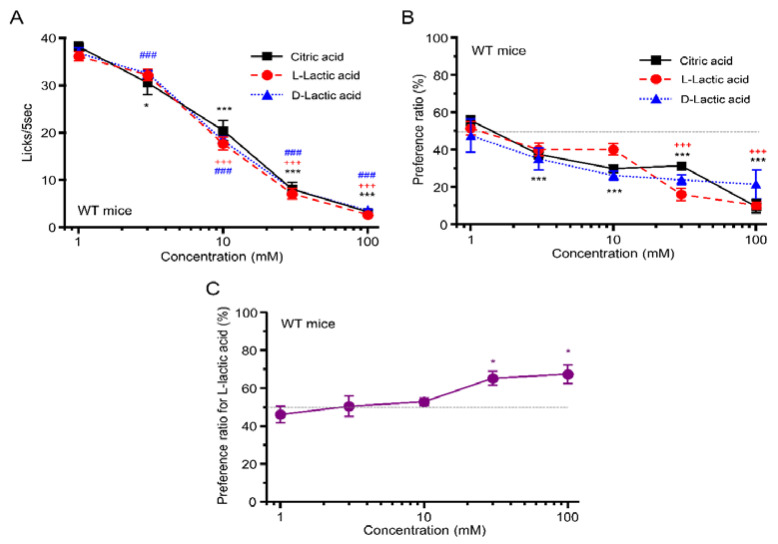
Behavioral responses to sour compounds in WT mice. (**A**) The number of licks of citric acid (black squares), l-lactic acid (red circles), and d-lactic acid (blue triangles) in the short-term (5 s) lick test (*n* = 7). (**B**) Preference ratio for citric acid (black squares), l-lactic acid (red circles), and d-lactic acid (blue triangles) in the 48 h two-bottle test (*n* = 6). (**C**) Preference ratio for l-lactic acid over d-lactic acid in the 48 h two-bottle test (*n* = 6). * *p* < 0.05, *** *p* < 0.001, ^+++^
*p* < 0.001, ^###^
*p* < 0.001, post hoc Tukey HSD test (vs. 1 mM). All data are presented as the mean ± standard error.

**Figure 4 foods-12-01150-f004:**
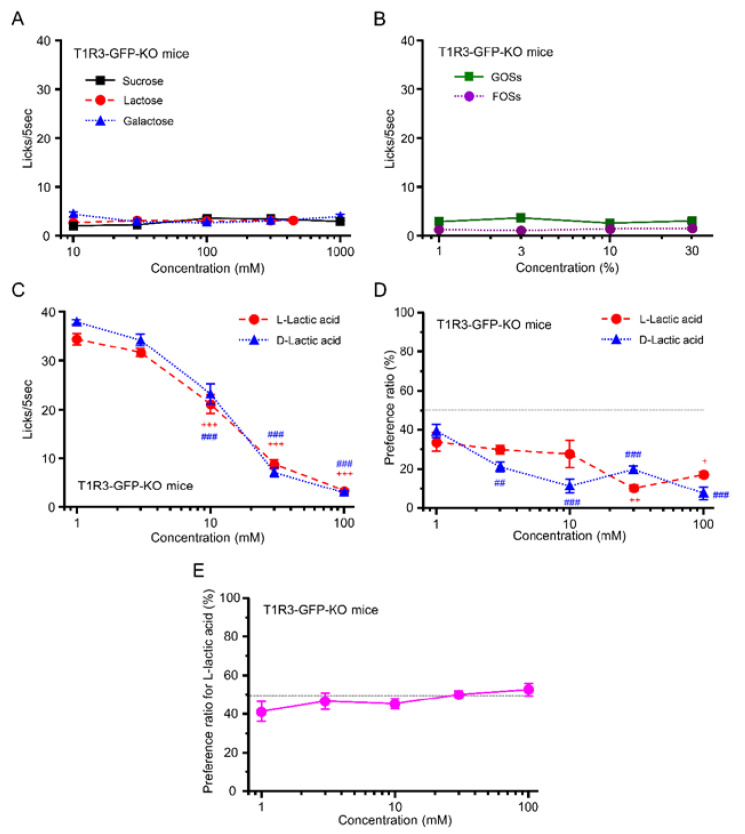
Behavioral responses to multiple ingredients in mice lacking T1R3 (T1R3-GFP-KO mice). (**A**) The number of licks of sucrose (black squares), lactose (red circles), and galactose (blue triangles) in the short-term (5 s) lick test (*n* = 8). (**B**) The number of licks of GOSs (green squares) and FOSs (purple circles) in the short-term (5 s) lick test (*n* = 8). (**C**) The number of licks of l-lactic acid (red circles) and d-lactic acid (blue triangles) in the short-term (5 s) lick test (*n* = 8). (**D**) Preference ratio for l-lactic acid (red circles) and d-lactic acid (blue triangles) in the 48 h two-bottle test (*n* = 6). (**E**) Preference ratio for l-lactic acid over d- lactic acid in the 48 h two-bottle test (*n* = 6). ^+^
*p* < 0.05, ^++^
*p* < 0.01, ^##^
*p* < 0.01, ^+++^
*p* < 0.001, ^###^
*p* < 0.001, post hoc Tukey HSD test (vs. 1 mM). All data are presented as the mean ± standard error.

**Figure 5 foods-12-01150-f005:**
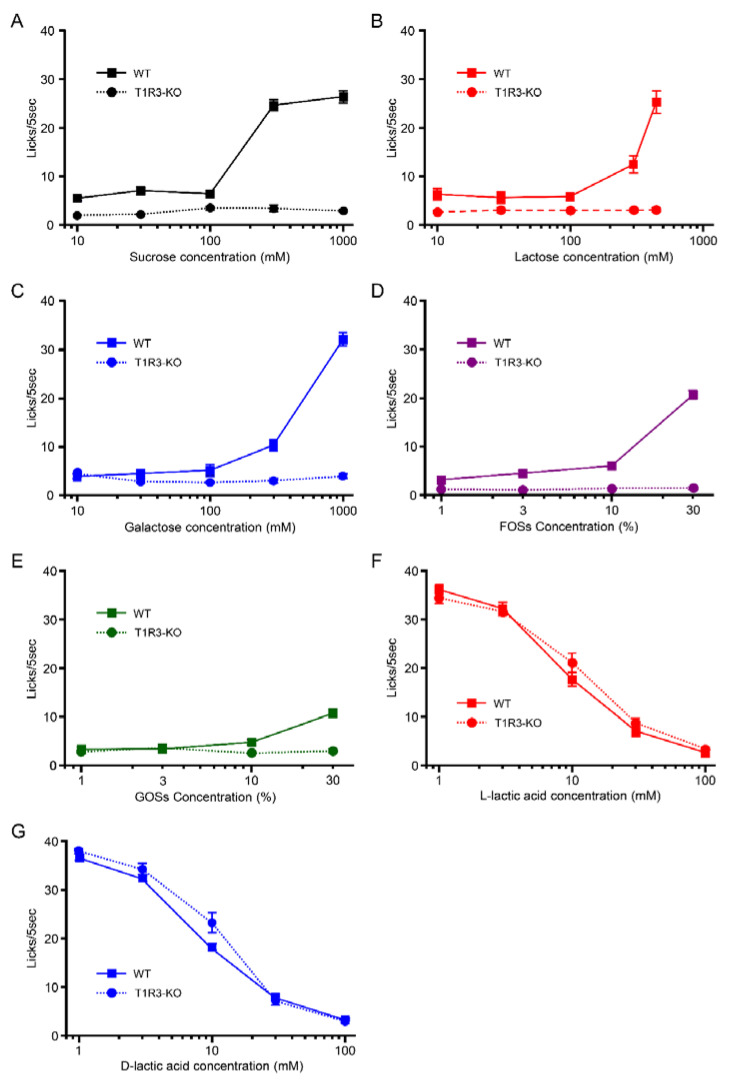
Comparison of short-term lick test between WT and T1R3-GFP-KO mice. (**A**) Sucrose. (**B**) Lactose. (**C**) Galactose. (**D**) FOSs. (**E**) GOSs. (**F**) l-lactic acids. (**G**) d-lactic acid. The WT mice (*n* = 7). The T1R3-GFP-KO mice (*n* = 8). All data are presented as the mean ± standard error. ANOVA results for these data are summarized in Table 1.

**Figure 6 foods-12-01150-f006:**
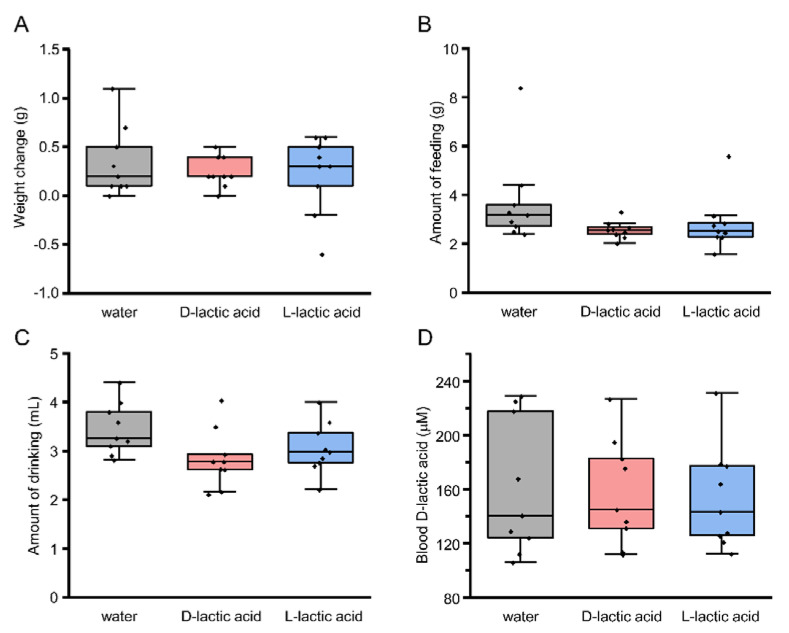
Blood d-lactic acid levels after ingestion of l- or d-lactic acid. (**A**–**C**) Weight change of mice (**A**), total food intake (**B**), and total solution intake (**C**) during 12 h presentation of food and solution (water, 30 mM l-lactic acid, or 30 mM d-lactic acid) (*n* = 9). (**D**) Blood d-lactic acid levels after 12 h presentation of food and solution (water, 30 mM l-lactic acid, or 30 mM d-lactic acid) (*n* = 9). In these box plots, the box indicates the 25th and 75th percentiles; the line across the box, the median; and whiskers, maximum and minimum values. Each point indicates individual data.

**Table 1 foods-12-01150-t001:** Two-way ANOVA results for short-term lick test.

Ingredient	Effect	Degree of Freedom	F Value	*p* Value
Sucrose	genotype	1.65	712	<0.001
	concentration	4.65	136	<0.001
	interaction	4.65	120	<0.001
Lactose	genotype	1.65	154	<0.001
	concentration	4.65	33.4	<0.001
	interaction	4.65	31.6	<0.001
Galactose	genotype	1.65	262	<0.001
	concentration	4.65	128	<0.001
	interaction	4.65	118	<0.001
GOSs	genotype	1.52	206	<0.001
	concentration	3.52	128	<0.001
	interaction	3.52	108	<0.001
FOSs	genotype	1.52	49.1	<0.001
	concentration	3.52	22.8	<0.001
	interaction	3.52	24.5	<0.001
l-lactic acid	genotype	1.65	1.04	0.31
	concentration	4.65	327	<0.001
	interaction	4.65	1.62	0.179
d-lactic acid	genotype	1.65	2.52	0.117
	concentration	4.65	452	<0.001
	interaction	4.65	2.68	0.039

## Data Availability

All data supporting the findings of this study are available from the corresponding author upon request.

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
