# Peer review of "Taste Responses and Ingestive Behaviors to Ingredients of Fermented Milk in Mice"

_foods, 2023, doi:10.3390/foods12061150_

Round 1

Reviewer 1 Report

Yamase et al. present an interesting study aiming to describe the synergic and isolated organoleptic properties of sour-tasting components of fermented milk. The study is fairly well-design and provides interesting insight into the understanding of the chemosensory response.

However, some points of concern should be raised.

Introduction:

- 4th paragraph, totally unnecessary.

- Lines 70-71. "Taste receptors are expressed throughout the body; for example, sweet taste receptors are involved in digestion and absorption" The authors provide functional examples to illustrate distribution. Please rephrase. 

Methods:

- Not clear why the use of the short-term and long-term licking paradigms. 

- Animals were deprived of water and food before exposure. Given the caloric value provided by sugars, how does the study control for motivational factors regardless of organoleptic preference? 

- The study fails to show relevant information on a control group with water compared to sugars.

Discussion:

- In lines 284-285, how do the results suggest a sweet/umami taste of L-lactic acid? This relationship is not clear. More argumentation is needed.

Author Response

Thank you very much for your helpful comments and suggestions. We have revised our manuscript to address your suggestions. We hope our revised manuscript will now be found satisfactory.

Reviewer’s comment:

Introduction:

- 4th paragraph, totally unnecessary.

Reply: Since there have been few taste studies related to lactose and oligosaccharides, we think this paragraph is necessary to give readers the prerequisite knowledge.

- Lines 70-71. "Taste receptors are expressed throughout the body; for example, sweet taste receptors are involved in digestion and absorption" The authors provide functional examples to illustrate distribution. Please rephrase.

Reply: As reviewer suggested, we revised this sentence to " Taste receptors are expressed throughout the body; for example, sweet taste receptors in the gastrointestinal tract are involved in enhancement of absorption of glucose [18] and bitter taste receptors function in biological defense in the trachea [19].".

Methods:

- Not clear why the use of the short-term and long-term licking paradigms.

Reply: As reviewer suggested, we added some descriptions in final paragraph of introduction “As a characteristic point, we used two-types of licking tests. One is short-term (5s) lick test to examine purely taste responses to each ingredient. Another is long term (48h) two bottle test, which involves post ingestive effect of each ingredient in addition to taste response.”.

- Animals were deprived of water and food before exposure. Given the caloric value provided by sugars, how does the study control for motivational factors regardless of organoleptic preference?

Reply: Because short term lick test is used to analyze oral "taste" aspect of stimulant, we do not consider about "caloric" aspect of sugars/saccharides. Food deprivation is used to enhance motivation to drink sweet solution, and well used procedure in short term lick test (such as Ma et al., Neuron, 2018)

- The study fails to show relevant information on a control group with water compared to sugars

Reply: Because quinine is added to all sugars/saccharides to obtain clear concentration responses to these compounds in short term lick test, we believe that comparison of water and sugars/saccharides+quinine may be meaningless.

Discussion:

- In lines 284-285, how do the results suggest a sweet/umami taste of L-lactic acid? This relationship is not clear. More argumentation is needed.

Reply: As reviewer suggested, we revised this part of Discussion as follows;

" WT mice showed similar avoidance to both L- and D-lactic acid—sour ingredients of fermented milk—in both short-term lick tests and long-term preference tests (Figs. 3A, 3B) but preferred L- over D-lactic acid in the long-term test (Fig. 3C). Although T1R3-GFP-KO mice also showed similar avoidance to both L- and D-lactic acid in both short-term lick tests and long-term preference tests (Fig. 4C,D), they did not prefer L- over D-lactic acid in the long-term test (Fig. 4E). These results suggest that L-lactic acid but not D-lactic acid may be detected by taste receptors containing T1R3, inducing a sweet or umami taste in addition to a sour taste."

Reviewer 2 Report

Fermented milk foods such as yogurts are highly prevalent in the modern diet, and may provide health benefits to humans. These foods are often described as sour tasting. The study by Yamase et al aims to fill an important gap in knowledge about the effects of taste and post-ingestive factors of various compound associated with fermented milk foods on intake behaviors. Although the study has the potential to reveal important new information, test designs, statistical analyses, and interpretation have limitations outlined below.

         1.       Several aspects of the short-term licking tests complicate interpretation.

 First, although the stated purpose of the experiments is to examine taste responses to fermented milk nutrients, such as sugar, quinine is added in mixture to each of these individual palatable components. The rationale for this is not clear, as these food products are not described to have a bitter taste. It is also unclear if quinine is added to the water stimulus presented in each test.

 Second, multiple tastant concentrations are presented for each stimulus, which is a nice aspect of the design, but they are presented in descending order of concentration (or ascending order for aversive tastants), rather than in a random order. Provided the mice are water and food restricted prior to some tests, and water deprived for others, changes in concentration-depending licking could be partially related to satiation across the test. The authors have not addressed this.

 Third, the number of short-term licking test trials is not provided.

 Fourth, for the T1R3 KO mice, licks to all sugar or saccharide concentrations are very low. This is surprising if the mice were water/food restricted. It is not clear if low lick numbers were also observed with water. If so, it could be that the concentration of quinine used masked or overshadowed any potential weak, but palatable, non-sweet/umami taste component. This should be discussed.

2. The reason that the primary statistical comparisons are focused on different solutions/concentrations within genotype is not in line with the stated research questions. Further, it is unclear how meaningful those types of comparisons/results are. 

First, more detail should be provided about the statistical approach in general.

Second, it is unclear why the authors opted to compare lick responses or preference among sugars, OS, and lactic acids by concentration. The more interesting and informative comparison would be to compare wild type mice to the T1R3 KO on each stimulus and/or concentration. This would help support any direct conclusions about the role of sweet/umami taste in preference for fermented milk ingredients. As it is, these conclusions can only be inferred. Moreover, the authors should acknowledge that calorie content and sugar concentration are not matched on galactose versus lactose and sucrose, for example.

   3. The inclusion of both short-term licking tests and long term intake tests within this study is a nice aspect of the design, and provides new information about the role of taste versus other factors in intake patterns. However, the authors conclude, including in the abstract, that differences in short term and long-term tests suggests postingestive factors, including conditioned taste aversion. The patterns of long-term intake are interesting, but the authors have not directly tested conditioned taste aversion here, and so any conclusions about that should be removed.

          4.  The authors tested whether D-lactic acidosis could be one of the factors contributing to decreased long term preference for D-lactic acid versus L-lactic acid. This was a nice aspect of the study, but it will also be important to consider if wild type mice are better able to metabolize L-lactic acid than T1R3 KO mice.

          5. Preference for either lactic acid was lower than for water at higher concentrations in both wild type and T1R3 KO mice. Are there intake differences in D- and/or L-lactic acid among each genotype? Further, short term licking tests that directly compare the two lactic acids could help to provide more information if this is related to taste.

Author Response

Thank you very much for your helpful comments and suggestions. We have revised our manuscript to address your suggestions. We hope our revised manuscript will now be found satisfactory.

Reviewer’s Comment-

1.Several aspects of the short-term licking tests complicate interpretation.

First, although the stated purpose of the experiments is to examine taste responses to fermented milk nutrients, such as sugar, quinine is added in mixture to each of these individual palatable components. The rationale for this is not clear, as these food products are not described to have a bitter taste. It is also unclear if quinine is added to the water stimulus presented in each test

Reply: The reason for adding quinine is to obtain clear concentration-dependent preference to sweeteners in the short-term lick tests. As cited in [22], the method of adding quinine has been used in previous papers. We think the description was not clear, so we have added the description in Methods section. We did not add quinine to water stimulus.

Second, multiple tastant concentrations are presented for each stimulus, which is a nice aspect of the design, but they are presented in descending order of concentration (or ascending order for aversive tastants), rather than in a random order. Provided the mice are water and food restricted prior to some tests, and water deprived for others, changes in concentration-depending licking could be partially related to satiation across the test. The authors have not addressed this

Reply: In Methods section, we missed some aspects of presentation order in previous MS. For preferable tastants, we presented in descending order of concentration in first trial then randomized order in second and further trials. For aversive tastants, we presented in ascending order of concentration in first trial then randomized order in second and further trials. This presentation order in first trial is important to obtain clear concentration dependent responses in short term lick test. These were mentioned in Methods section of revised MS. Food deprivation may be important to obtain clear concentration dependent responses to sweeteners. Food deprivation may enhance motivation to drink preferable solutions. Similar strategy was used in previous study (such as Ma et al., Neuron, 2018).

Third, the number of short-term licking test trials is not provided.

Reply: The number of short-term licking test trials was at least three. It is stated in the short-term lick test part of the Methods, but we apologize for the lack of clarity in the description.

Fourth, for the T1R3 KO mice, licks to all sugar or saccharide concentrations are very low. This is surprising if the mice were water/food restricted. It is not clear if low lick numbers were also observed with water. If so, it could be that the concentration of quinine used masked or overshadowed any potential weak, but palatable, non-sweet/umami taste component. This should be discussed.

Reply: T1R3-KO mice, which lack sweet/umami taste receptors, are not able to sense sweet. Therefore, T1R3-KO mice lost preference to sweeteners in short term lick test because short term lick test has been used to examine taste sensitivity of animals. In our method, quinine was mixed with sugars or saccharides, therefore, T1R3-KO mice might only sense the bitter taste when they drink bitter-sugar/saccharide mixture. Thus T1R3-KO mice did not lick such mixture if sugar/saccharide concentration is very high. Of course, it is possible that T1R3-KO mice may sense weak palatable components in such mixture. However, it is difficult to reveal such weak preference (maybe sweet taste) in short term lick test. We added data for lick to water in short term lick test in Result section of revised MS. T1R3-KO mice lick water well during examination of sugars/saccharides+quinine.

2. The reason that the primary statistical comparisons are focused on different solutions/concentrations within genotype is not in line with the stated research questions. Further, it is unclear how meaningful those types of comparisons/results are.

First, more detail should be provided about the statistical approach in general.

Reply: We revised Statistical analysis in Methods section as follows;

"For short-term lick tests and long-term, 48-hour 2-bottle tests, differences among concentrations of each ingredient were statistically analyzed by one-way ANOVA and post hoc Tukey highest-significant-difference (HSD) test. Differences among taste compounds (GOSs vs FOSs, D-lactic acid vs L-lactic acid) and concentration were statistically analyzed by two-way ANOVA. Differences among species (WT vs T1R3-GFP-KO) and concentration were statistically analyzed by two-way ANOVA. Differences in blood D-lactic acid levels, weight changes, and amount of feeding and drinking were statistically analyzed by one-way ANOVA."

Second, it is unclear why the authors opted to compare lick responses or preference among sugars, OS, and lactic acids by concentration. The more interesting and informative comparison would be to compare wild type mice to the T1R3 KO on each stimulus and/or concentration. This would help support any direct conclusions about the role of sweet/umami taste in preference for fermented milk ingredients. As it is, these conclusions can only be inferred. Moreover, the authors should acknowledge that calorie content and sugar concentration are not matched on galactose versus lactose and sucrose, for example.

Reply: As reviewer suggested, we added comparison between WT and T1R3-GFP-KO mice in Results of revised MS. Because of lack of T1R3, sensitivity to sugars and oligosaccharide, which have a strong sweet taste at higher concentration, may be different between WT and T1R3-GFP-KO mice. We also acknowledged in Result section that calorie content and sugar concentration are not matched on galactose versus lactose and sucrose.

3. The inclusion of both short-term licking tests and long term intake tests within this study is a nice aspect of the design, and provides new information about the role of taste versus other factors in intake patterns. However, the authors conclude, including in the abstract, that differences in short term and long-term tests suggests postingestive factors, including conditioned taste aversion. The patterns of long-term intake are interesting, but the authors have not directly tested conditioned taste aversion here, and so any conclusions about that should be removed

Reply: As reviewer suggested, we did not perform any experiments on the conditioned taste aversion. In revised manuscript, we removed description about conditioned taste aversion.

4. The authors tested whether D-lactic acidosis could be one of the factors contributing to decreased long term preference for D-lactic acid versus L-lactic acid. This was a nice aspect of the study, but it will also be important to consider if wild type mice are better able to metabolize L-lactic acid than T1R3 KO mice.

Reply: As reviewer suggested, we added some discussion about metabolism of L-lactic acid in Discussion section of revised manuscript as follows;

"Third possibility is that wild type mice may be better able to metabolize L-lactic acid than T1R3-GFP-KO mice. Sweet taste receptors are known to be involved in digestion, and absorption in the intestine [18]. Such sweet receptors in the gastrointestinal tract may be involved in the metabolism of L-lactic acid, inducing some positive postingestive effect in WT mice but not in T1R3-GFP-KO mice. Such possibility also should be investigated in future studies."

5. Preference for either lactic acid was lower than for water at higher concentrations in both wild type and T1R3 KO mice. Are there intake differences in D- and/or L-lactic acid among each genotype? Further, short term licking tests that directly compare the two lactic acids could help to provide more information if this is related to taste.

Reply: There is no intake data in this paper, but we did not include the data because there was no significant difference in D-and/or L-lactic acid intake among each genotype.  Direct comparison of D- and L- lactic acid in short term lick test was shown in this paper and there was no statistical difference between them. However, it is difficult to show the slight difference of taste of two compounds by short-term lick test. Other technique such as gustatory nerve recording may be useful to elucidate such difference between taste compounds. But this is beyond of this MS.

Round 2

Reviewer 1 Report

I feel satisfied with the authors' response to my inquiries.

Author Response

Thank you.

Reviewer 2 Report

The authors have addressed many of the prior issues raised by reviewers, and more adequate details regarding the Method and Statistical Analyses are now included. Furthermore, while this reviewer appreciates that the genotype comparisons are now included in this revised manuscript, the overall conclusions, interpretation, and impact of the data would be better served if this wildtype versus T1R3 KO comparison were presented in the main figures for each taste solution or test. As it stands, the comparison among different solutions/concentrations within each genotype is not as meaningful. These could be relegated to a table, if necessary. Second and more minor, in the Introduction (e.g., lines 76-79) and elsewhere (e.g., section title for 2.4), the authors refer to the 48 hour preference test as a lick test. This is incorrect. In these two bottle preference tests, intake was measured. It is inappropriate to suggest this is a measure of licking behavior, and these could reflect very different processes. 

Author Response

Thank you very much again for your helpful comments and suggestions. We have revised our manuscript to address your suggestions. We hope our revised manuscript will now be found satisfactory.

The authors have addressed many of the prior issues raised by reviewers, and more adequate details regarding the Method and Statistical Analyses are now included. Furthermore, while this reviewer appreciates that the genotype comparisons are now included in this revised manuscript, the overall conclusions, interpretation, and impact of the data would be better served if this wildtype versus T1R3 KO comparison were presented in the main figures for each taste solution or test. As it stands, the comparison among different solutions/concentrations within each genotype is not as meaningful. These could be relegated to a table, if necessary.

Reply: As reviewer suggested, we added figure showing comparison between WT mice and T1R3-KO mice in revised Fig. 5.

Second and more minor, in the Introduction (e.g., lines 76-79) and elsewhere (e.g., section title for 2.4), the authors refer to the 48 hour preference test as a lick test. This is incorrect. In these two bottle preference tests, intake was measured. It is inappropriate to suggest this is a measure of licking behavior, and these could reflect very different processes. 

Reply: We agree with reviewer's suggestion. We changed "licking " to "intake" in appropriate points of revised MS.